# Fully Automatic Whole-Volume Tumor Segmentation in Cervical Cancer

**DOI:** 10.3390/cancers14102372

**Published:** 2022-05-11

**Authors:** Erlend Hodneland, Satheshkumar Kaliyugarasan, Kari Strøno Wagner-Larsen, Njål Lura, Erling Andersen, Hauke Bartsch, Noeska Smit, Mari Kyllesø Halle, Camilla Krakstad, Alexander Selvikvåg Lundervold, Ingfrid Salvesen Haldorsen

**Affiliations:** 1Mohn Medical Imaging and Visualization Centre, Department of Radiology, Haukeland University Hospital, 5009 Bergen, Norway; skka@hvl.no (S.K.); kari.strono.wagner-larsen@helse-bergen.no (K.S.W.-L.); njal.gjerde.lura@helse-bergen.no (N.L.); erling.andersen@helse-bergen.no (E.A.); hauke.bartsch@helse-bergen.no (H.B.); noeska.smit@uib.no (N.S.); alexander.selvikvag.lundervold@hvl.no (A.S.L.); 2Department of Mathematics, University of Bergen, 5020 Bergen, Norway; 3Department of Computer Science, Electrical Engineering and Mathematical Sciences, Western Norway University of Applied Sciences, 5063 Bergen, Norway; 4Section of Radiology, Department of Clinical Medicine, University of Bergen, 5021 Bergen, Norway; 5Department of Clinical Engineering, Haukeland University Hospital, 5021 Bergen, Norway; 6Department of Informatics, University of Bergen, 5020 Bergen, Norway; 7Department of Obstetrics and Gynecology, Haukeland University Hospital, 5053 Bergen, Norway; mari.halle@uib.no (M.K.H.); camilla.krakstad@uib.no (C.K.); 8Centre for Cancer Biomarkers, Department of Clinical Science, University of Bergen, 5021 Bergen, Norway

**Keywords:** cervical cancer, deep learning, tumor segmentation

## Abstract

**Simple Summary:**

Uterine cervical cancer (CC) is a leading cause of cancer-related deaths in women worldwide. Pelvic magnetic resonance imaging (MRI) allows the assessment of local tumor extent and guides the choice of primary treatment. MRI tumor segmentation enables whole-volume radiomic tumor profiling, which is potentially useful for prognostication and individualization of therapy in CC. Manual tumor segmentation is, however, labor intensive and thus not part of routine clinical workflow. In the current work, we trained a deep learning (DL) algorithm to automatically segment the primary tumor in CC patients. Although the achieved segmentation performance of the trained DL algorithm is slightly lower than that for human experts, it is still relatively good. This study suggests that automated MRI primary tumor segmentations by DL algorithms without any human interaction is possible in patients with CC.

**Abstract:**

Uterine cervical cancer (CC) is the most common gynecologic malignancy worldwide. Whole-volume radiomic profiling from pelvic MRI may yield prognostic markers for tailoring treatment in CC. However, radiomic profiling relies on manual tumor segmentation which is unfeasible in the clinic. We present a fully automatic method for the 3D segmentation of primary CC lesions using state-of-the-art deep learning (DL) techniques. In 131 CC patients, the primary tumor was manually segmented on T2-weighted MRI by two radiologists (R1, R2). Patients were separated into a train/validation (*n* = 105) and a test- (*n* = 26) cohort. The segmentation performance of the DL algorithm compared with R1/R2 was assessed with Dice coefficients (DSCs) and Hausdorff distances (HDs) in the test cohort. The trained DL network retrieved whole-volume tumor segmentations yielding median DSCs of 0.60 and 0.58 for DL compared with R1 (DL-R1) and R2 (DL-R2), respectively, whereas DSC for R1-R2 was 0.78. Agreement for primary tumor volumes was excellent between raters (R1-R2: intraclass correlation coefficient (ICC) = 0.93), but lower for the DL algorithm and the raters (DL-R1: ICC = 0.43; DL-R2: ICC = 0.44). The developed DL algorithm enables the automated estimation of tumor size and primary CC tumor segmentation. However, segmentation agreement between raters is better than that between DL algorithm and raters.

## 1. Introduction

Uterine cervical cancer is one of the leading causes of cancer-related deaths in women, particularly in developing countries [1]. For local staging, magnetic resonance imaging (MRI) is the preferred imaging modality due to its high soft-tissue resolution, conspicuously depicting the tumor and its boundaries to the surrounding tissue. Routine diagnostic work-up at many centers includes multiparametric MRI with diffusion weighted imaging (DWI), allowing the assessment of local tumor extent and maximum tumor diameter.

MRI radiomic tumor profiling involves the extraction of quantitative imaging information from segmented tumor masks using mathematical descriptors [2]. Radiomic tumor profiles have been linked to clinical phenotypes and prognosis for several cancers. Currently, there is a growing body of literature suggesting that the radiomic profile in CC is associated with prognostic factors [3,4,5], and predicts therapeutic response [6] and outcome [7,8,9].

Accurate tumor segmentation is a critical step in radiomic profiling since the radiomic data is specifically extracted from the segmented tumor volumes. Manual tumor segmentation in 3D by experts is, however, very labor intensive, making it unfeasible in routine clinical practice. Thus, a seamless clinical integration of whole-volume radiomic tumor profiling requires the development of robust platforms for accurate automated tumor segmentation. Previous CC studies applying deep learning (DL) networks for automated primary tumor segmentation on MRI data report highly variable Dice scores (Dice scores: 0.44-0.93) between DL segmentation and tumor segmentation derived by radiologists [10,11,12,13]. Furthermore, poor reproducibility of certain radiomic parameters derived from automatic tumor segmentations have been reported [12].

Traditional methods for medical image segmentation have relied upon techniques such as thresholding, edge detection, region-growing, clustering, or they have been based on the evolution of partial differential equations. However, over the past decade, DL-based segmentation methods have been shown to outperform classical segmentation methods, and have become state-of-the-art for complex segmentation tasks [14,15,16,17]. A common approach is to use models based on the U-Net architecture [18], which is an encoder-decoder convolutional neural network (CNN). U-Net-based models have been successfully employed in a wide range of medical imaging applications, including multi-parametric MRI tumor segmentation. The segmentation algorithm applied in the present work is an enhanced residual U-Net model [19].

This study aimed to use state-of-the-art DL libraries for automated CC segmentation. By training the platform on multiparametric pelvic MRI data in patients diagnosed with uterine CC we aimed to evaluate a DL algorithm for automated primary tumor segmentation in CC.

## 2. Methods

### 2.1. MRI Acquisitions

A total of 135 uterine CC patients diagnosed during 2009–2017 who underwent pretreatment pelvic MRI (including DWI) and had visible tumors at MRI when assessed by two radiologists (hereafter referred to as Rater 1 and 2) were included in this study. Two of the patients were excluded due to poor image quality, and two patients with very large primary tumors (> 1000 mL) were excluded, since more than two patients would be needed to be able to train a robust model on very large tumors. Thus, a total of 131 CC patients comprised the final study cohort.

The MRI examinations consisted of T2-weighted sequences and DWI with either two, three or four b-values. Apparent diffusion coefficient (ADC) maps were generated from mono-exponential fits to the DWI, using vendor-provided software at the scanner. T2-weighted images, high b-value images and ADC-maps were all available when the raters manually segmented whole-volume tumor masks on the T2-weighted images. The MRI examinations were performed at multiple hospitals using different MRI scanners and protocols (see Table 1 for details).

The three imaging channels (T2-weighted, high b-value and ADC maps) were subsequently used in separate data sets for training (*n* = 90), validation (*n* = 15) and testing (*n* = 26) of the hlDL segmentation network.

### 2.2. Inclusion Criteria

This retrospective study was conducted under Institutional Review Board (IRB)-approved protocols (2015/2333/REK vest) with written informed consent from all patients at primary diagnosis. All patients were diagnosed and treated at Haukeland University Hospital, Bergen, Norway. A total of 131 patients with histologically verified uterine cervical cancer who underwent pretreatment MRI between 2009 and 2017 were included. The patients were selected from a larger CC patient cohort scanned during 2002–2017 based on the following inclusion criteria for imaging data: (i) visible tumor on pelvic MRI confirmed by both radiologists; (ii) axial/axial oblique (relative to the long axis of the cervix) T2-weighted images; and (iii) axial/axial oblique DWI. An overview of patient characteristics in the training/validation and test cohorts is provided in Table 2.

### 2.3. Manual Tumor Segmentation

We used the open-source software ITK-SNAP (v. 3.6.0; www.itksnap.org, accessed on 16 June 2020) [20] for manual 3D tumor segmentation. The primary uterine cervical tumor was manually segmented on T2-weighted images, using axial oblique (when available) or axial images. Segmentations were performed by one radiologist in 105 patients (Rater 1 [K.W.L.]: *n* = 58; Rater 2 [N.L.]: *n* = 47) or both radiologists in 26 patients (comprising the test cohort). Rater 1 (R1) and Rater 2 (R2) had 12 and 7 years of experience in reading pelvic MRI, respectively. The radiologists were blinded to clinicopathologic patient information but had the DWI images available to support placement of tumor segmentations. The extracted 3D image mask was exported in the Neuroimaging Informatics Technology Initiative (NIfTI) file format [21].

### 2.4. Major Processing Steps

A flow chart of major processing steps is illustrated in Figure 1. The train and validation cohort comprised 105 randomly chosen patients used for training and validation of the 3D U-Net. Within this cohort, 90/105 sets went into training by stratified sampling based on tumor volume. The remaining 15/105 went into the validation data set and were used for reporting validation parameters during training. For a total of three times during the development period, the training and validation cohorts were selected at random from the set of 105 patients in order to increase the robustness of the algorithm and to avoid over-optimistic or over-pessimistic test results. The test cohort comprised 26 patients who had primary tumors manually segmented by both raters, serving as an unbiased test set for the evaluation of segmentation performance and inter-rater agreement.

A sigmoid transformation was applied to the activation map compiled in the DL algorithm, providing a smooth function between zero and one. A final binary model prediction was derived by thresholding this function at a value of 0.5 [22]. However, thresholding leads to a binary map potentially containing multiple objects. In order to select the most probable mask object representing the primary tumor, we computed mean activation values within each individual object in the binary model prediction, with a neighborhood stencil size of 3. The object with the highest mean activation value was automatically chosen to represent the primary tumor. The activation map superimposed on a T2-weighted image of one patient is shown in Figure 2, generated using a 3D Slicer [23]. In this example, two potential tumor objects were identified. The object with the largest mean activation value was finally selected as a primary tumor. Segmentation performance was assessed in terms of DL-based tumor volumes and tumor masks’ location compared with the R1 and R2 tumor segmentations.

### 2.5. Evaluation of Segmentation Performance

To compare segmentation performance metrics, we used the Dice-Sørensen similarity coefficient (DSC) [24], measuring the degree of regional overlap between two segmentations. We also used the Hausdorff distance (HD) as a measure of maximum distance between segmented contours, as this is more sensitive to outliers in the segmentation shape, not sufficiently captured by DSC. The parameters are defined as
$$\begin{array}{c} \mathrm{DSC}=\frac{2|X\cap Y|}{\left|X|+|Y\right|},\;\mathrm{HD}\left(x,y\right)=\max\left\{\delta (x,y),\delta (x,y)\right\} \end{array}$$
where |·| is the cardinality and $\delta \left(x,y\right):=\sup_{x\in X}\inf_{y\in Y}d\left(x,y\right)$[25] for the Euclidean distance d(x,y) between *x* and *y*. As a metric for segmentation performance, we also compare the estimated tumor volume between the hlDL algorithm and R1/R2.

The comparison of segmentation performance between R1 and R2 is referred to as inter-rater agreement. Similarly, the comparison of performance between the DL algorithm and R1 and R2 is referred to as DL-R1 and DL-R2, respectively. Median DSC and HD reported in Table 3 were adjusted to inter-rater agreement according to the formulas DSC(DL,R1/R2) ← DSC(DL,R1/R2) + (1-DSC(R1,R2) and HD(DL, R1/R2) ← HD(DL,R1/R2) − HD(R1,R2).

Patient characteristics for the training/validation and test data sets were compared using the Mann–Whitney U test for continuous variables and Pearson’s chi-square test (*n* > 5 in any group) or Fisher exact test (*n* ≤ 5 in any group) for categorical variables. Differences in median DSC and HD between DL and raters (DL-R1/R2), and between raters (R1-R2) were assessed using the Wilcoxon rank test. The agreement for primary tumor volumes between raters (R1-R2) and between DL and raters (DL-R1/R2) was reported using Bland–Altman plots and intraclass correlation coefficient (ICC). The difference in median tumor volume between raters and DL was assessed using Friedman’s test. Correlations between tumor volumes and DSC or HD were tested using the Spearman correlation with H${}_{0}$ of zero Spearman’s ρ. Multiple linear regression was used to investigate statistical relations between T2- and DWI field-of-view (FOV) (defined as FOV${}_{x}\times$ FOV${}_{y}$), T2- and DWI anisotropy (defined as slice thickness/max(pixel spacing x, pixel spacing y)), and field strength (1.5T or 3.0T) with DSC as the response variable. *p*-values below 0.05 are considered statistically significant. The statistical analyses were carried out in MATLAB using the Statistics and the Machine Learning Toolbox Version 12.0 (R2020b).

### 2.6. Implementation Details

Data sets were manipulated using the open-source, Python-based package Imagedata [26] for the reading and writing of image data between DICOM (https://dicomstandard.org, (accessed on 1 March 2018)) or NIfTI file format and NumPy arrays [27]. An in-house developed algorithm applying the geometric coordinate transformation specified within the DICOM image header was used for spatial alignment of the DWI data (ADC map and high *b*-value image) with the T2-weighted image using trilinear interpolation. After transformation, image voxel data for each patient was specified on the same spatial grid. Out-of-grid extrapolation values were set to zero.

We implemented our segmentation model using the 3D U-Net architecture from the MONAI framework (https://monai.io,[19]], accessed on 17 December 2021), with 5 layers of 16, 32, 64, 128, 256 channels, respectively, each with downsampling and upsampling by a factor of 2, and a skip connection between them. The training was performed using our own extension of the fastai library [28,29], which simplifies training of three-dimensional convolutional neural networks using modern best-practices for training deep neural networks.

Before training, all MRI volumes were resampled to isotropic (0.7×0.7×0.7) mm${}^{3}$ voxel size using RegularGridInterpolator from SciPy [30]. The interpolation method was ‘trilinear’ for the MRI data, and ‘nearest neighbor’ for the binary mask data. They were channel-wise normalized using z-normalization (i.e., zero mean and unit standard deviation), and resized to 304 × 304 × 144 dimensions using either cropping or zero padding. The image data were of different matrix sizes, and the amount of cropping and padding was therefore different between data sets. We used data parallelism in PyTorch to train our model, a batch size of 4, and trained the model for 60 epochs using Dice loss function [31] on four NVIDIA Tesla V100 32 GB GPUs. We employed a Ranger optimizer [32] with an initial learning rate of 0.1, rapidly decreasing during the final few epochs using a cosine annealing scheduler, an idea that is related to the concept of super-convergence [33]. For data augmentation, we used random zooming by a factor in the range [1, 1.2], and random elastic deformations with 5 control points along each dimension of the coarse grid with a maximum displacement set f to 4 along each direction at each control point. The transformations were performed on the fly during training, with a probability set to 0.2 for each transformation. The weights of our final model were selected based on a callback that monitored the DSC on the validation data after each epoch, with the condition of saving the model if the performance of the validation data was improved by at least 0.005 × DSC from the currently best model. Source code used in this work is openly available via GitHub (https://github.com/MMIV-DL/cervical-cancer-segmentation-2022, accessed on 21 March 2022).

## 3. Results

### 3.1. Train and Validation Metrics

Train and validation losses, as well as the DSC of the validation data set, are reported in Figure 3 as a function of epoch number. The train loss is steadily decreasing, indicating numerical stability of the optimization algorithm. Epoch number 55 (highlighted with bold in the figure) represented the optimal stopping point, when the validation DSC reaches a plateau and before it starts decreasing due to the effect of over-training. This approach minimizes the risk of over-training, potentially lowering general performance on unseen data sets. The Dice score in the validation data set reached a value of 0.52 when using this optimal stopping point (Epoch number 55) (Figure 3).

A histogram depicting the distribution of predicted objects in the test cohort (*n* = 26) is shown in Figure 4. The median (min, max) number of objects per patient was 7 (1.28). Only 12% (3/26) of the patients had one DL mask object, by definition representing the predicted primary tumor. For the vast majority (88%; 23/26), the predicted mask image contained multiple objects. For these patients, the object expressing the highest mean activation value was automatically selected to represent primary tumor.

### 3.2. Performance in Terms of DSC and HD

A summary of segmentation performance metrics in terms of DSC and HD for DL- and R1/R2 segmented primary tumor masks is given in Table 3. Segmentation performance of the DL algorithm is lower than that for the raters both in terms of median DSC (DL-R1: DSC = 0.60, DL-R2: DSC = 0.58, R1-R2: DSC = 0.78; Wilcoxon rank sum test, *p* ≤ 0.01) and median HD (DL-R1: HD = 29.2 mm; DL-R2: HD = 30.2 mm, HD = 14.6 mm; Wilcoxon rank sum test, *p* ≤ 0.01).

Box plots of DSC and HD for DL segmentation compared with that of R1 and R2 are depicted in Figure 5, reflecting the reported performance values in Table 3, I.

After adjusting for R1-R2 disagreement, estimates of segmentation performance are significantly higher both in terms of median DSCs (DL-R1: DSC = 0.81, DL-R2: DSC = 0.79) and median HDs (DL-R1: HD = 3.73 mm, DL-R2: HD = 9.10 mm).

### 3.3. Performance in Terms of Reported Tumor Volume

Bland–Altman plots comparing reported primary tumor volumes based on segmentations by DL and R1/R2 and R1 and R2 are shown in Figure 6. The mean difference in tumor volume between DL-R1/R2 and R1-R2 was low for all comparisons (≤0.94 mL), suggesting high agreement in mean tumor volume. However, higher LoA of ±60/75 mL were found for DL-R1/R2 compared to R1-R2 with LoA of ±24 mL. There was no difference in median DL/R1/R2 tumor volumes (Friedman’s test, *p* = 0.10). Agreement in terms of ICCs for log tumor volume for DL-R1 and DL-R2 was lower (ICC${}_{\mathrm{DL},R1}$ = 0.43 with 95% CI = (0.07, 0.70), *p* = 0.01, and ICC${}_{\mathrm{DL},R2}$ = 0.44 with 95% CI = (0.08, 0.70), *p* = 0.01, respectively) than that for R1-R2 (ICC${}_{R1,R2}$ = 0.93 with 95% CI = (0.85, 0.97), *p* < 0.001).

A relatively weak but significant dependency of tumor volume on segmentation performance was observed for the DL algorithm (DL-R1: ρ = 0.40, *p* = 0.046; DL-R2: ρ = 0.41, *p* = 0.039, Spearman rank correlation) (Figure 7, upper row, left and middle panel). For R1-R2, large tumor size only tended to be positively correlated with segmentation performance (R1-R2: ρ = 0.31, *p* = 0.12; Spearman rank correlation) (Figure 7, upper row, right panel). All plots suggest a log-like relationship between increasing tumor size and DSC. We found no significant correlation between tumor volume and HD (ρ≤ 0.24, *p* ≥ 0.24, Spearman rank correlation) (Figure 7, lower row). Patients with a low DSC < 0.2 and a small tumor volume <50 mL are pairwise indicated in the upper and lower rows. For the ML-R1/R2 relation (Figure 7, left and middle columns), these cases (*n* = 6) had large HDs, whereas for the R1-R2 relation, similar cases (*n* = 2) had relatively low HDs (Figure 7, right column).

The multiple linear regression model reported in Table 4 revealed no linear relationship between field strength, T2/DWI anisotropy, and T2/DWI FOV as explanatory variables and DSC as response variable.

## 4. Discussion

Patients diagnosed with uterine cervical cancer (CC) in high-income countries routinely undergo imaging by pelvic MRI, allowing the assessment of primary tumor extent and tumor invasion to surrounding tissue or pelvic lymph nodes. MRI-based whole-volume radiomic tumor profiling is promising for prognostication [3,4,5] and tailoring of cancer treatment [6,7,8,9]. However, the clinical utility of CC radiomic profiling is hampered by labor intensive manual tumor segmentations. In the current work based on 131 manually segmented primary CC lesions, we present a deep learning based algorithm for tumor segmentation yielding a fully automatic prediction of primary tumor position and boundaries.

This DL-based fully automatic approach for primary CC segmentations yielded relatively high segmentation performance (DL-R1: median DSC = 0.60, DL-R2: DSC = 0.58), although still lower than that for the expert raters (R1-R2: DSC = 0.78) (Table 3). Importantly, with a DSC of 0.78 for R1-R2, it is evident that substantial disagreement also exists when human experts define primary tumor boundaries in CC. Without using consensus segmentations across multiple raters, it seems inherently impossible to train a DL algorithm to yield better segmentation performance than that achieved for the raters involved in training the model. Thus, in an attempt to adjust for disagreement across raters we also report adjusted DSCs and HDs for the DL segmentation (Table 3). These adjusted performance metrics (DL-R1/R2: DSC = 0.81/0.79; HD = 3.73/9.10) are as expected better than the corresponding crude estimates (DL-R1/R2: DSC = 0.60/0.58; HD = 29.2/30.2) and may be considered as relatively good.

In the present study, the crude performance estimates are lower than that reported in some previous studies of CC tumor segmentation [10,11,12,13]. Bnouni et al. reported a DSC of 0.93 (using T2-weighed MRI) [13] (*n* = 15), Kano et al. reported a DSC score of 0.83 (using diffusion-weighted MRI) [11] (*n* = 98), and Lin et al. reported a DSC score of 0.82 (using multiparametric MRI) [12] (*n* = 169). However, these studies all used k-fold cross-validation applied to a train/validation data set for performance estimation and hyperparameter tuning. This setup is unfortunately not directly comparable to the present study since they did not estimate the performance of their DL algorithm in a separate and unbiased test data set [34].

Lin et al. presented a DL algorithm for automated tumor segmentations in CC using T2-weighted 3T MRI with DWI [12] (*n* = 169). Similar to our study, they used a separate test set to assess the performance of their DL algorithm, and report a DSC of 0.82. However, Lin et al. did not report inter-rater agreement as their manual tumor segmentations used for training of the DL network were by a single radiologist, however, with subsequent verification by a second radiologist. Thus, although the crude performance estimates of our DL algorithm (median DSCs of 0.60/0.58) seems inferior to that of the DL algorithm by Lin et al. (DSC of 0.82), the adjusted performance estimates for our DL algorithm (DSCs of 0.81/0.79) are quite comparable to that of their DL algorithm.

Interestingly, recent studies presenting DL algorithms for automated MRI tumor segmentations of other pelvic malignancies report performance metrics with DSCs in the range of 0.52–0.84 [35,36,37,38,39], i.e., prostate cancer (DSC of 0.52 using k-fold cross-validation [35] [*n* = 204]), endometrial cancer (DSC of 0.77/0.84 using a test set [36] [*n* = 139] and DSC of 0.81 using k-fold cross-validation [37] [*n* = 200]), and rectal cancer (DSC of 0.68/0.70 using a test set [38] [*n* = 140] and DSC of 0.70 using a test set [39] [*n* = 300]). Hence, our DSCs for the DL algorithm in CC (DL-R1: median DSC = 0.60, DL-R2: DSC = 0.58) are quite comparable to that of other pelvic malignancies. Similarly, inter-rater agreement in our study (R1-R2: DSC = 0.78) compares well with that reported in prostate cancer (DSC of 0.57, *n* = 78) [40] and rectal cancer (DSC of 0.83, *n* = 140) [38].

The use of one or multiple raters for the annotation of data sets may influence the segmentation performance of the DL algorithm. Interestingly, the study of Ji et al. [41] report higher performance of the DL algorithm when the algorithm utilizes the rich annotation information derived from manual segmentations from multiple raters. We included annotated data sets from two raters in the training data, however, with segmentations by a single rater for each patient in the training/validation set. Future work in CC segmentation should explore the value of rigorously incorporating information from multiple raters in order to maximize performance [41,42].

A further possible reason for variable segmentation performance for DL algorithms may be related to the extent of harmonization of input data. Although previous studies have identified dependencies of image resolution and noise characteristics on the reproducibility of radiomic features [43], we found no direct association between FOV, voxel anisotropy, and field strength on segmentation performance (cfr. Table 4). Still, the importance of using homogeneous imaging data in terms of standardized MR protocols for successfully training and applying a DL algorithm is not fully known. Despite z-normalization of the data prior to feeding the algorithm, variations in site, hardware and acquisition parameters in our study may have influenced the data in a way that has increased the complexity of the segmentation task. On the contrary, it is also possible that the algorithm becomes more robust by being exposed to variation in the training process, and that this may increase the performance of the algorithm when faced with new challenging segmentation tasks on images acquired at different sites and MRI scanners [44].

A majority of the raw predicted mask images contained multiple separate objects in 3D (23/26), with many of these being outside the uterine cervix. Although some of these additional objects could potentially represent extrauterine tumor tissue or metastases, they could not by definition represent primary tumor, and most of these objects were due to noise. A commonly used approach to handle multiple output regions is to threshold the predicted mask objects based on expected size [36,45], often in combination with various morphological operations [46]. We pursued an automatic approach that selected the most probable mask object based on maximum value of the average activation within the mask object. This approach attempts to take advantage of the inherent certainty built into the hlDL network, being expressed as high activation values whenever the network has high certainty for a tumor prediction, and oppositely expressing low activation values in the presence of low certainty.

The estimated tumor volume revealed no difference in median values between DL/R1/R2 (Friedman test, *p* = 0.10, and Bland–Altman plots, Figure 6). However, larger LoA for DL-R1/R2 compared to that of R1/R2 and higher ICC for R1-R2 (ICC = 0.93) than for DL-R1/R2 (ICC = 0.43/0.44) suggests that human experts reach a higher accuracy for tumor segmentation than the DL network. Future work must clarify how this observed difference in segmentation accuracy may influence radiomic feature extraction and potential prognostic modeling from corresponding radiomic signatures.

Interestingly, there was a positive, significant correlation between DSC and tumor volume for DL-R1/R2 (ρ≥ 0.40, *p* ≤ 0.046) but only a tendency for R1-R2 (ρ = 0.31, *p* = 0.12) (Figure 7). This finding indicates that accurate tumor segmentation by the DL algorithm or even by human experts is easier to achieve if the tumors are relatively large. Our findings further support that whenever the DL method is failing in the presence of small tumors, the DL-suggested tumor mask is either very large or located far from the cervical lesion (Figure 7). Cases with inter-rater disagreement for small tumors had masks that were closer in space. This observed difference is probably because human experts have been trained to differentiate the uterine cervix from other organs.

Notably, similar findings with better segmentation accuracy in larger tumors have been reported in CC [12] (*n* = 169), and brain tumors [47] (*n* = 69). Identifying small tumors in CC can be a challenging task also for trained raters due to lack of contrast and inherent difficulties in distinguishing between normal and pathological tissue. Still, the weak relationship observed suggests that the challenges in retrieving accurate manual and automatic CC segmentations are only partly related to tumor size.

This study has several limitations. Our imaging data were acquired at different scanners with large variations in field-of-view, pixel size and field strength, leaving many images prone to substantial padding-/cropping effects and resizing in the preprocessing steps prior to feeding the network with data. The use of more standardized imaging protocols would potentially reduce the need for post-processing steps that are known to reduce data quality. However, it may be argued that this setup using imaging data derived from different scanners with their variable protocols, more truly reflects the standard imaging work-up that CC patients in general undergo. Furthermore, we excluded two patients with a tumor size > 1000 mL due to this small number being insufficient to train a model for large tumors. Thus, our findings in terms of performance in relation to tumor size may not be extrapolated to patients with extremely large tumors.

In conclusion, we have developed a DL algorithm for fully automatic primary tumor segmentation in CC that yields highly promising segmentation performance, although not yet reaching the same segmentation performance as human raters. With likely breakthroughs in DL technologies in the near future, this should motivate further development of similar DL platforms to enable automated radiomic tumor profiling in CC.

## Figures and Tables

**Figure 1 cancers-14-02372-f001:**
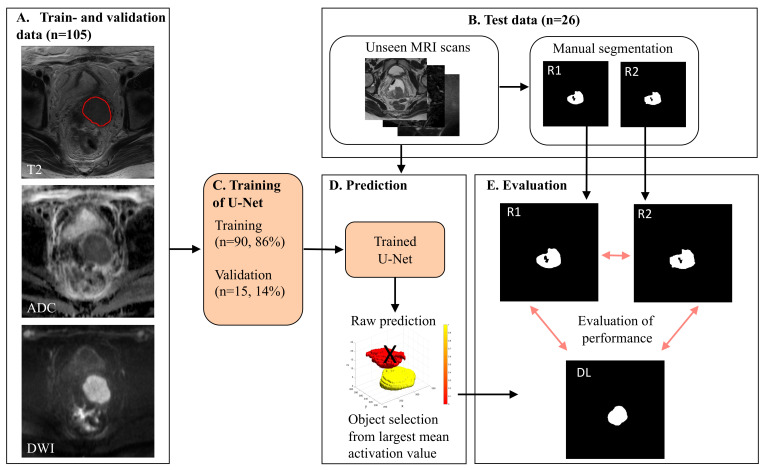
Graphical illustration of DL (deep learning) workflow and study setup. MRI data included T2-weighted images and diffusion weighted images (DWI), using high b-value images and apparent diffusion coefficient (ADC) maps at primary diagnostic work-up in 131 CC patients. (**A**) In the train and validation cohort (*n* = 105), primary tumor was segmented by one of the two expert raters (R1: *n* = 58, R2: *n* = 47). (**B**) The test cohort (*n* = 26), with primary tumor segmentations by both expert raters (R1 and R2), served as an unbiased test set for evaluating performance of the DL algorithm and inter-rater agreement. (**C**) The train and validation cohort (*n* = 105) was used to train a 3D U-Net using 90/105 (86%) cases for training and 15/105 (14%) cases for validation. (**D**) The trained network predicted raw tumor masks in the test data set (*n* = 26), identifying multiple regions in 23/26 cases. The object with the largest mean activation value was selected as primary tumor (yellow object). Other objects with lower mean activation values (red object) were removed from further analysis (indicated by a black cross). (**E**) DL-derived tumor masks were compared with manually segmented masks from R1 and R2, using Dice score and Hausdorff distances. R1 = Rater 1, R2 = Rater 2.

**Figure 2 cancers-14-02372-f002:**
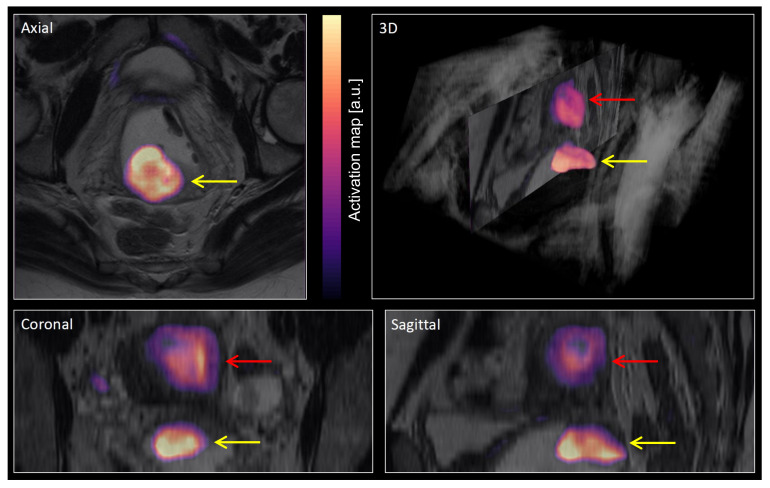
A visualization of the activation map (colored regions) from the DL (deep learning) segmentation superimposed on T2-weighted MRI (grayscale colormap) for three orthogonal planes and using 3D volume rendering. The activation map was later transformed with a sigmoid function and then thresholded, resulting in a binary prediction map. Two objects were identified in this patient: The object positioned in the uterine cervix (yellow arrows) had the largest mean activation value and was thus automatically selected to represent primary tumor. The object positioned in the uterine cavity/body (red arrows) had lower mean activation value and was thus excluded.

**Figure 3 cancers-14-02372-f003:**
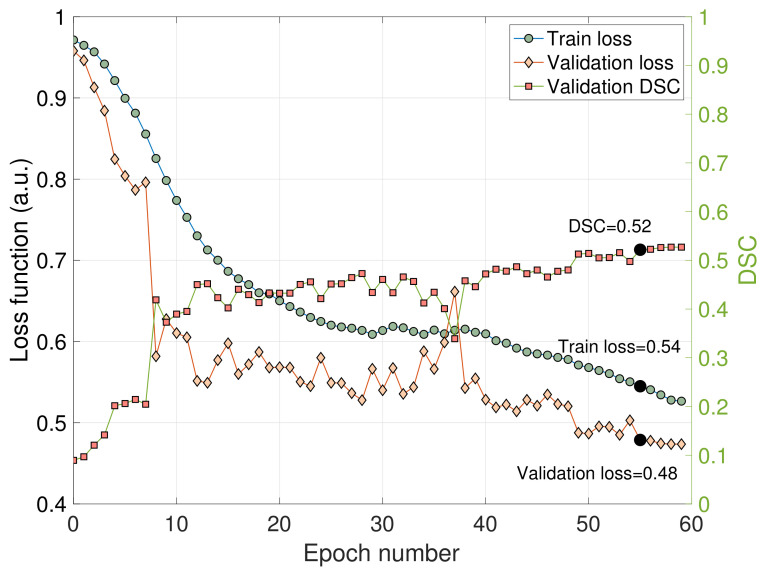
Train and validation losses (left axis) and Dice scores (DSC) (right axis) depicted as a function of epoch number. The train loss is smoothly decreasing, indicating numerical stability of the algorithm. The Dice score reaches a plateau, suggesting an optimal epoch number of 55 (black, solid dots). This epoch number yields optimal training performance of the network while minimizing the risk of over-training. a.u. = arbitrary units.

**Figure 4 cancers-14-02372-f004:**
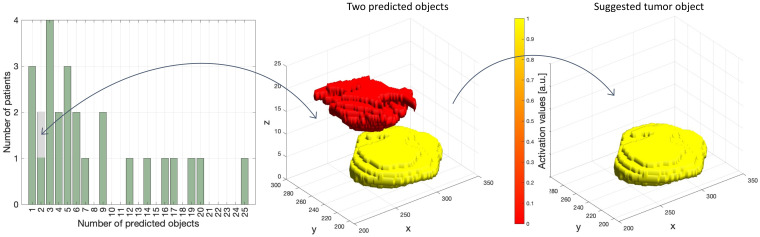
(**Left**): Histogram depicting number of objects in the prediction maps for the test cohort (*n* = 26) using a sigmoid-transformed activation map with a threshold of 0.5. In only 3/26 patients a single object was identified, whereas in 23/26 patients multiple mask objects were suggested. (**Middle**): Surface rendering depicting two objects (in red and yellow) in one of the patients having two predicted objects (grey box in histogram). The surface colors red/yellow indicate corresponding low/high mean activation values for the two objects (a.u. = arbitrarily units). (**Right**): In this patient with two suggested objects, the yellow mask with highest mean activation value was automatically selected as primary tumor.

**Figure 5 cancers-14-02372-f005:**
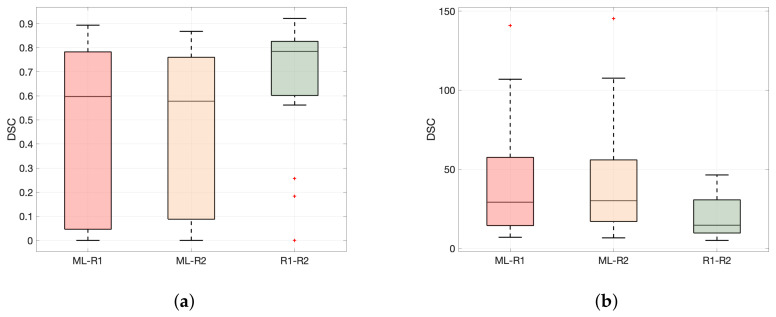
Comparison of (**a**) Median Dice coefficient (DSC) and (**b**) Median Hausdorff distance (HD) for segmentations by DL-R1, DL-R2, and R1-R2. Agreement between R1-R2 is significantly better than between DL and R1/R2 in terms of DSC and HD (Wilcoxon rank sum test, *p* ≤ 0.01). The central line indicates the median, and the upper and edges of the box indicate the 25th and 75th percentiles, respectively. The whiskers indicate the most extreme data points not considered to be outliers, while outliers are plotted individually using a ‘+’ symbol. R1 = rater 1; R2 = rater 2.

**Figure 6 cancers-14-02372-f006:**
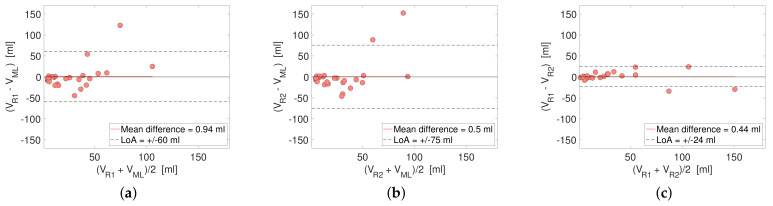
Bland–Altman plots comparing tumor volumes V [mL] from (**a**) DL (deep learning) and R1, (**b**) DL and R2 and (**c**) R1 and R2. Red lines indicate mean difference of the estimate, and dashed lines represent lower and upper limits-of-agreement (LoA). Mean difference in estimated tumor volumes is low for all comparisons, indicating a high agreement in mean primary tumor volume by all methods. However, LoA is higher for DL-R1/R2 than for R1-R2, indicating a higher individual disagreement for tumor measurements by DL-R1/R2 than by R1-R2. R1 = rater 1; R2 = rater 2.

**Figure 7 cancers-14-02372-f007:**
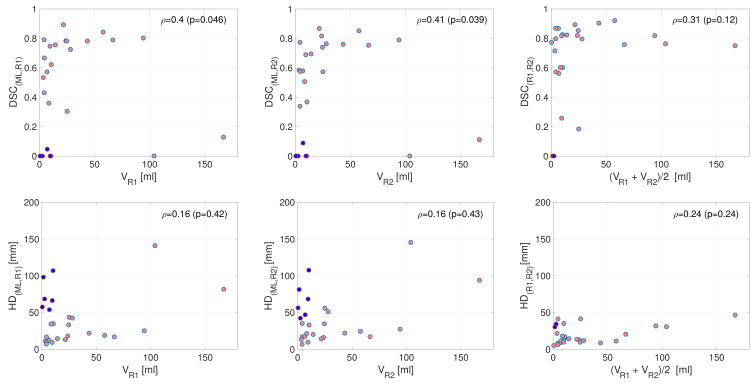
Tumor volume in relation to segmentation performance. (**Left**): R1 tumor volume. (**Middle**): R2 tumor volume. (**Right**): Mean tumor volume for R1- and R2 masks. (**Upper row**): Tumor volume against Dice coefficient (DSC). There is a weak but significant correlation between primary tumor volume and DSC for DL (deep learning)-R1/R2 (left and middle panel, *p* ≤ 0.046). R1-R2 DSC only tended to be associated with tumor volume (right panel, *p* = 0.12). (**Lower row**): Plots of tumor volume against Hausdorff distance (HD). We found no significant correlation between primary tumor volume and HD for any of the associations DL-R1/R2 or R1-R2 (*p*≥ 0.24). Both rows: The same patients with (i) low DSC < 0.2 and (ii) a small tumor volume <50 mL (estimated tumor volume for this condition is either R1 (left), R2 (middle), or mean (R1, R2) tumor volume) are simultaneously marked in blue in upper and lower panels, suggesting that patients experiencing a low DSC are normally high in HD for ML-R1/R2 (left and middle panels, the same *n* = 6 patients were identified). For R1-R2, patients with low DSC also have low HD (right panel, *n* = 2 patients). R1 = rater 1; R2 = rater 2; V = tumor volume, {ρ, *p*} = Spearman rank correlation coefficient with associated *p*-value.

**Table 1 cancers-14-02372-t001:** Summary of MRI protocols used in the study cohort (*n* = 131). The MRI data were acquired using different protocols, field strength and vendors. T2-weighted and diffusion-weighted imaging (DWI) acquisition parameters are reported as median values. FA = flip angle; FOV = field of view; mm = millimeters; ms = milliseconds; NA = not available; *n* = Number of patients in each category in terms of field-strength and vendor; s = seconds; T2 = T2-weighted; T = Tesla; TE = echo time; TR = repetition time. * Available b-values are reported, but not all b-values were available after export of the image data from the scanner.

	Parameter	Siemens 1.5T	GE 1.5T	Philips 1.5	Siemens 3T	Philips 3T
T2	Pixel spacing [mm] (inplane)	(0.39, 0.39)	(0.35, 0.35)	(0.40, 0.40)	(0.52, 0.52)	(0.35, 0.35)
	Matrix (x, y)	(512, 512)	(512, 512)	(512, 512)	(384, 384)	(512, 512)
	FOV [mm] (x, y)	(180, 180)	(180, 180)	(205, 205)	(200, 200)	(180, 180)
	TR [ms]	4790	3157	5362	4610	4074
	TE [ms]	100	81	100	94	110
	FA [degrees]	150	160	90	148	90
	Slice thickness [mm]	3.00	3.00	3.00	3.00	2.50
	Number of averages	2	2	6	2	2
	Interslice gap [mm]	0.50	0.00	0.30	0.30	0.25
	Number of slices	25	30	26	24	35
DWI	Pixel spacing [mm] (x, y)	(1.56, 1.56)	(1.37, 1.37)	(1.46, 1.46)	(1.43, 1.43)	(0.80, 0.80)
	Matrix (x, y)	(144, 144)	(256, 256)	(256, 256)	(144, 144)	(352, 352)
	FOV [mm] (x, y)	(250, 250)	(350, 350)	(375, 375)	(200, 200)	(280, 280)
	TR [ms]	3200	4000	1716.30	5640	3280
	TE [ms]	82	52	69.18	63	85
	FA [degrees]	90	90	90	180	90
	Slice thickness [mm]	4.00	5.00	5.00	3.00	4.00
	Number of averages	10	2	3	2	2
	Interslice gap [mm]	0.60	0.50	1.00	0.40	0.40
	Number of slices	22	25	30	25	33
	*b*-values [s/mm${}^{2}$]	[0/50, 800/1000]	NA *	[0, 1000]	[0/50, 800/1000]	NA *
N	Number of patients	51	9	27	27	9

**Table 2 cancers-14-02372-t002:** Patient characteristics of the training/validation cohort (*n* = 105) and the test cohort (*n* = 26). The two patient cohorts have similar clinicopathological characteristics. ${}^{1}$ Mann–Whitney U test. ${}^{2}$ Pearson’s chi-square test. ${}^{3}$ Fisher exact test. ${}^{4}$
*n* = 97 for training/validation cohort and *n* = 25 for the test cohort. * Adenosquamous, neuroendocrine, and undifferentiated carcinomas; FIGO = International Federation of Gynecology and Obstetrics; IQR = Interquartile range; w/o = with and without.

Variable	Train (*n* = 90) and Validation (*n* = 15) Data	Test Data (*n* = 26)	*p*
Age (yrs.)			0.73 ${}^{1}$
Median (IQR)	48 (37–60)	49 (41–59)	
FIGO (2009) stage			0.21 ${}^{2}$
I	52 (49%)	14 (54%)	
II	27 (26%)	6 (23%)	
III	18 (17%)	5 (19%)	
IV	8 (8%)	1 (4%)	
MRI-assessed maximum tumor size (cm)			0.24 ${}^{1}$
Median (IQR)	4.6 (3.0–5.6)	3.9 (2.5–5.1)	
Primary treatment			0.21 ${}^{2}$
Surgery only	26 (25%)	9 (34%)	
Surgery and adjuvant therapy	63 (60%)	15 (58%)	
Primary radiotherapy w/o chemotherapy	12 (11%)	2 (8%)	
Palliative treatment	4 (4%)	0	
Histologic subtype			0.19 ${}^{2}$
Squamous cell carcinoma	82 (78%)	21 (81%)	
Adenocarcinoma	18 (17%)	3 (11%)	
Other *	5 (5%)	2 (8%)	
Histologic grade ${}^{4}$			0.76 ${}^{3}$
Low/medium	80 (82%)	22 (88%)	
High	17 (18%)	3 (12%)	

**Table 3 cancers-14-02372-t003:** Median (IQR = interquartile range) Dice score (DSC) and Hausdorff distance (HD) for tumor masks derived from DL (deep learning) segmentation compared to manual tumor segmentations by R1/R2. I: DL yields tumor masks with lower DSC and higher HD for DL-R1/R2 than that for R1-R2 (Wilcoxon rank sum, *p* ≤ 0.01 and *p*≤ 0.01, respectively). II: Performance metrics of tumor masks after adjusting for median R1-R2 disagreement. The adjusted values yield higher DSCs and lower HDs for DL-R1/R2 when using DSC = 1 and HD = 0 as reference values for R1-R2 (ref. values). * Statistically significant; ${}^{1}$ Statistical testing and difference in estimates do not change from I to II; R1 = rater 1; R2 = rater 2.

	Measure	Median Value of Estimate (IQR)			Absolute Difference (*p*-Value)	
		A **. (DL, R1)**	B **. (DL, R2)**	C **. (R1, R2)**	|A−C| **(** * **p** * **)**	|B−C| **(** * **p** * **)**
I. Unadjusted	DSC	0.60 (0.05, 0.78)	0.58 (0.09, 0.76)	0.78 (0.60, 0.83)	0.19 (0.01 *)	0.21 (0.005 *)
	HD [mm]	29.2 (14.5, 57.5)	30.2 (17.1, 55.9)	14.6 (9.80, 30.7)	14.6 (0.01 *)	15.5 (0.003 *)
II. Adjusted for R1-R2 disagreement	DSC	0.81	0.79	1 (ref.)	- ${}^{1}$	- ${}^{1}$
	HD [mm]	3.73	9.10	0 (ref.)	- ${}^{1}$	- ${}^{1}$

**Table 4 cancers-14-02372-t004:** Association between field strength, anisotropy T2 and DWI, field-of-view (FOV) T2 and DWI, and DSC using multiple linear regression. None for the MRI acquisition features had a statistical assocation to segmentation performance (*p* ≥ 0.33, multiple linear regression).

	Estimate	SE	*p*
(Intercept)	0.08	0.36	0.82
Field strength	−0.06	0.15	0.71
Anisotropy T2	0.04	0.04	0.33
FOV T2	1.77	23.69	0.94
Anisotropy DWI	0.04	0.07	0.54
FOV DWI	4.10	6.11	0.51

## Data Availability

The Institutional Review Board approval did not allow for openly sharing of the MRI datasets. Source codes used for data processing and analyses are available in the Github repository https://github.com/MMIV-DL/cervical-cancer-segmentation-2022, accessed on 21 March 2022.

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
