# Peer review of "Fully Automatic Whole-Volume Tumor Segmentation in Cervical Cancer"

_cancers, 2022, doi:10.3390/cancers14102372_

Round 1
Reviewer 1 Report
The authors proposed an algorithm to perform whole tumor segmentation in cervical cancer. By using similar number of 3D tumor segmentation provided by 2 radiologists, the authors trained 3D U-Net for tumor segmentation. There are some major parts to revise in this manuscript.
First of all, 3D U-Net is a type of convolutional neural network which is generally considered for deep learning, not for machine learning. Such claim in the manuscript must be corrected. Secondly, it is well known that 3D U-Net architecture is quite similar to the U-Net and considered as old network structure. This brings low originality for deep learning algorithm. Furthermore, the DSCs reported in this manuscript is considered significantly lower than Reference 12 provided in the manuscript, while U-Net is used as algorithm for tumor segmentation in cervical cancer in Reference 12 published in 2019. Although the authors provided the adjusted performance estimated for their algorithm, there is no description provided by the authors how to perform the adjustment between the physician 1 and physician 2’s manual tumor segmentation. Therefore, it is hard to compare authors’ results and the results provided from Reference 12.
Hence, major revision is required for this manuscript. The authors should consider improving their network structure and robustness of their model for publication.
Reviewer 2 Report
This study aimed to use state-of-the-art deep learning libraries to develop a ML platform for automated cervical cancer (CC) segmentation. By training this platform on multiparametric pelvic MRI data in patients diagnosed with uterine CC we aimed to develop a ML algorithm for automated primary tumor segmentation in CC. The paper looks good with respect to automatic segmentation. But repeatability of the features from the scanner data is not tested. Intra observal variability and ground truth need to be focus more in details. Deep learning may produce variability based on the ground truth, so I would suggest STAPLE method for ground truth generation. How does the segmentation method impact for the small tumor is not very clear to me.
You can get some insight from the following papers:
1) Roy S, Whitehead TD, Li S, Ademuyiwa FO, Wahl RL, Dehdashti F, Shoghi KI. Co-clinical FDG-PET radiomic signature in predicting response to neoadjuvant chemotherapy in triple-negative breast cancer. Eur J Nucl Med Mol Imaging. 2022 Jan;49(2):550-562. doi: 10.1007/s00259-021-05489-8. Epub 2021 Jul 30. Erratum in: Eur J Nucl Med Mol Imaging. 2021 Sep 22;: PMID: 34328530; PMCID: PMC8800941. 2) Roy S, Whitehead TD, Quirk JD, Salter A, Ademuyiwa FO, Li S, An H, Shoghi KI. Optimal co-clinical radiomics: Sensitivity of radiomic features to tumour volume, image noise and resolution in co-clinical T1-weighted and T2-weighted magnetic resonance imaging. EBioMedicine. 2020 Sep;59:102963. doi: 10.1016/j.ebiom.2020.102963. Epub 2020 Sep 2. PMID: 32891051; PMCID: PMC7479492.Author Response
Please see attachment.

Round 2
Reviewer 1 Report
The revised version looks suitable for publication.
Reviewer 2 Report
Authors have solved all the issues raised by me.